# Persistence of Juvenile Idiopathic Arthritis-Associated Uveitis in Adulthood: A Retrospective Study

**DOI:** 10.3390/jcm11092471

**Published:** 2022-04-28

**Authors:** Maria Pia Paroli, Alessandro Abbouda, Giuseppe Albanese, Massimo Accorinti, Alessandro Falcione, Leopoldo Spadea, Marino Paroli

**Affiliations:** 1Department of Sense Organs, Eye Clinic, Sapienza University of Rome, 00161 Rome, Italy; a.abbouda@gmail.com (A.A.); alb.comgiu90@gmail.com (G.A.); massimo.accorinti@tiscali.it (M.A.); niggo89@hotmail.it (A.F.); leopoldo.spadea@uniroma1.it (L.S.); 2Division of Clinical Immunology, Department of Clinical, Anestesiologic and Cardiovascular Sciences, Sapienza University of Rome, 00161 Rome, Italy; marino.paroli@uniroma1.it

**Keywords:** JIA-associated uveitis, JIA in adulthood, uveitis complications, uveitis treatment

## Abstract

Background: Juvenile idiopathic arthritis (JIA) is a rheumatic condition of childhood that is frequently associated with anterior chronic uveitis. Evidence suggests that uveitis may persist up to adulthood in some cases, possibly causing severe visual impairment. Methods: We conducted a retrospective study on a series of patients aged 16 years or older with JIA-related active uveitis who were referred to the Uveitis Service of Sapienza University of Rome from 1990 to 2019 to evaluate the characteristics of ocular disease in patients with JIA-associated uveitis (JIA-U) who still exhibit uveitis in adulthood. Data on clinical features, treatment, complications and visual outcomes were collected. Results: Twenty adults (85% female; median age 23.4 ± 6.6 years, range 16–38 years) with ongoing uveitis (35 eyes) were identified. The median age at JIA onset was 6.15 ± 2.9 years (range 2–10), and uveitis onset was 8.7 ± 4.7 years (range 3–20). The patients were observed in a median follow-up of 16 ± 7.7 years (range 4–35). Fifty-seven percent of affected eyes (20 eyes) had good visual acuity (>0.4 logMAR), while eleven percent of affected eyes (4 eyes) were blind (≤20/200). Uveitis required topical steroids and mydriatic/cycloplegic in all cases. Orbital steroid injection was performed in 13 eyes. Systemic corticosteroids and biologic drugs were used in 14 patients. Conclusions: Although the visual prognosis of JIA-U has improved in recent years, persistent uveitis up to adulthood is still observed. Therefore, protracted follow-up of JIA-U patients is warranted because of the high burden of delayed visual complications.

## 1. Introduction

Juvenile idiopathic arthritis (JIA) is a heterogeneous group of diseases characterized by arthritis of unknown origin affecting the child population. Uveitis is the most common extra-articular manifestation of JIA, occurring in about a quarter of patients. Moreover, JIA is the leading cause of anterior uveitis in Europe and North America in children aged 16 years or younger [1,2,3]. JIA-associated uveitis (JIA-U) is an anterior uveitis with chronic and non-granulomatous features; it is often bilateral and asymptomatic. Uveitis is frequent in the JIA oligoarticular subtype [4], in females, in early-onset arthritis and in JIA associated with serum antinuclear antibodies (ANA) [1].

The median age at uveitis onset in patients with JIA is 6 years. Ocular manifestations generally develop within 5–7 years after the onset of joint disease but may occur up to 28 years after the development of arthritis. JIA-U may be silent or develop in children who have not started to exhibit joint disease yet.

JIA-U can be a sight-threatening condition. The severity of inflammation can lead to ocular complications like macular edema, epiretinal membrane, macular hole, glaucoma, ciliary body atrophy and inflammatory or ischemic neuropathy. Cataract may be associated with chronic uveitis, often due to corticosteroid therapy, usually requiring surgery [5,6,7,8,9,10].

Although JIA-U typically resolves during childhood, many clinical observations have shown that a significant number of patients with JIA show active uveitis also in adulthood. These patients commonly have severe forms of uveitis, often requiring intensive treatment. The aim of this study was to evaluate clinical features and visual prognosis in a patient cohort with JIA-U in which uveitis persisted even during adult age.

## 2. Materials and Methods

We conducted a retrospective analysis of 20 patients with JIA-U (35 eyes) in a series of patients aged ≥16 years old referred to the Ocular Immunovirology Service of Sapienza University of Rome from June 1990 to July 2019. The demographic characteristics of the studied population are reported in Table 1. The study was undertaken in accordance with the ethical principles of the Declaration of Helsinki. Institutional review board approval was also obtained. Inclusion criteria were a diagnosis of JIA-U according to the International League of Association for Rheumatology criteria. Ophthalmological examinations included a best-corrected visual acuity (BCVA) using Snellen charts, intraocular pressure assessment using Goldmann applanation tonometer, slit-lamp biomicroscopy and fundus examination using bilateral indirect ophthalmoscope. Each patient underwent routine laboratory examinations. According to the clinical history and the characteristics of uveitis, serologic and immunologic tests, including detection of autoantibodies or antibodies to infective agents and human leukocyte antigen (HLA) class-I and class-II class typing, were performed. Instrumental examinations included fluorescein and indocyanine green angiography, ultrasonography, optical coherence tomography (OCT), electrophysiologic tests and visual field tests. Changes in uncorrected distance visual acuity (logMAR) were defined as visual impairment when at least 0.4 and as legal blindness when 0.9 or more. Complications included posterior synechiae, band keratopathy, cataract, ocular hypertension or hypotension, macular edema and epiretinal membrane. Cataract was defined as the presence of 1 + nuclear sclerosis or 1 + cortical change or trace posterior subcapsular changes, according to Woreta et al. [11]. Macular edema was defined as central macular thickness (CMT) greater than 300 µm with or without cyst formation as detected by clinical examination and/or by spectral-domain OCT [12]. Ocular hypertension was defined as intraocular pressure elevation >21 mmHg, and hypotony was defined as an intraocular pressure < 5 mmHg. The presence of band keratopathy and posterior synechiae were diagnosed by slit-lamp examination. Epiretinal membrane formation, papillitis, retinal vasculitis or exudative retinal detachment were diagnosed by indirect ophthalmoscopy and instrumental examinations (fluorescein and indocyanine green angiography, ultrasonography, OCT). Best-corrected visual acuity was recorded at the beginning and end of the follow-up. Statistical analysis was performed using non-parametric tests using the GraphPad Prism software.

## 3. Results

Twenty patients (17 female and 3 male, ratio 5.6:1) with a median age of 23.4 ± 6.6 years (range 16–38 years) with ongoing uveitis (35 eyes) were included in the study. The median age was 6.15 ± 2.9 years at JIA onset (range 2–10) and 8.7 ± 4.7 years at uveitis onset (range 3–20). The median time between the beginning of arthritis and of uveitis was 1.3 ± 4.4 years (range 5–10). Patients were followed up for a median of 16 ± 7.7 years (range 4–35). Seventeen patients (85%) were ANA positive. Uveitis was anterior and chronic in all the examined subjects. Ocular involvement was bilateral in 15 cases (75%). Ongoing arthritis was observed in 14 (70%) patients. Oligoarthritis was the most frequent subtype observed, being present in 19 patients (95%), whereas systemic arthritis was observed in 1 case only.

Some of these patients were already in arthritis remission. The main demographic and clinical characteristics of the study population are summarized in Table 1.

Ocular complications were reported in Table 2. They were observed in 22 eyes (62.8%) of 12 patients. The most common complications were cataract (18 eyes; 52%) with onset at a median age of 21.5 ± 4.3, band keratopathy (18 eyes; 52%) at the median age of 18.9 ± 2.6 and posterior synechiae (17 eyes (48%) at the median age of 23.2 ± 3.4. Six patients underwent cataract surgery with intraocular lens (IOL) implantation (median age of 25.7 ± 5.6), and two patients also required glaucoma surgery. The incidence of posterior synechiae during the follow-up was 0.009/eye-year (EY) in JIA patients without posterior synechiae at baseline. The rate of band keratopathy, elevated IOP and cystoid macular edema was 0.006/EY. The incidence of newly diagnosed cataract was 0.01/EY, whereas the rate of hypotony, papillitis and epiretinal membrane was 0.002/EY. Exudative retinal detachment was 0.001/EY. These data have been reported in Table 2.

The visual prognosis was as follows: 57% of affected eyes (20 eyes) had good visual acuity (>0.4 logMAR), while 11% of affected eyes (4 eyes) were blind (≤20/200). There were no significant variations in visual acuity from the beginning to the end of the follow-up. When comparing patients (6 patients/11 eyes) in whom uveitis had lasted less than 15 years with those in whom it lasted more than 15 years, a better prognosis in the first group was observed. In particular, all cases in the first group had good visual acuity (>0.4 logMAR), while only sixty percent of the second group had good visual acuity. The difference was statistically significant (*p* = 0.02).

Uveitis required topical dexamethasone and mydriatic/cycloplegic in all cases. Orbital depot methylprednisolone injection was performed in 13 eyes. Topical medication to reduce intraocular pressure was used in 18 eyes, and two patients required the association with systemic medication to treat intraocular pressure. The arthritis treatment required NSAIDs, systemic prednisone, intra-articular steroids and immunosuppressants.

Conventional synthetic DMARDs (csDMARDs) were used in 8 patients while biologics including infliximab (40%), adalimumab (15%), rituximab (5%) and abatacept (5%) were used in 13 patients (Table 3). We did not find any difference in the visual prognosis of patients treated with csDMARDs alone or in combination with biologics.

## 4. Discussion

In this study, we analyzed a cohort of 20 patients diagnosed with JIA-U during childhood and presenting with uveitis up to the age of 16 years or older. Most patients were female, with a female to male ratio of about 6:1. This ratio was lower in comparison with other studies where the female to male ratio was 10:1 or higher [13,14]. This may reflect the geographic variations in JIA-U prevalence as previously reported [3].

We found that the median age of patients affected by JIA-U in their adulthood was 23 years old. Arthritis was still active in some of these patients. More than half of the subjects included in this study experienced a uveitis duration of 15 years or longer. This finding confirms similar observations from other studies, challenging the belief that JIA-U goes into remission during adolescence [4,10,15,16].

We found that the most frequent complications in patients who were not in remission in the adult age were the presence of posterior synechiae and the presence of cataract. Lens opacity can be accelerated by the presence of synechiae of the iris and topical and/or systemic steroid therapy. Cataract represents the main cause of visual acuity loss in 19 (81%) of cases, as reported in other studies [15,17]. Moreover, the opacity of the dioptric media is of particular relevance since it can lead to amblyopia in younger children [1]. A potential limitation of our cataract grading was that it was not based on the Lens Opacities Classification System (LOCS) III. This is a method designed for classifying lens opacities that involves a standardized classification of nuclear color at the slit lamp and cortical and posterior subcapsular cataracts at the slit lamp with retro-illumination [18]. However, LOCS III was introduced in 1993, and our study analyzed patients since 1990. It is desirable that further studies on JIA-U might take advantage of LOCS III to assess lens opacity complications better.

Other studies have reported that posterior complications are not uncommon in these patients. The most frequent are vitritis and cystoid macular edema, which represent important causes of decreased visual function with a frequency of up to 40%. Papillitis, epiretinal membranes, retinal vasculitis and exudative retinal detachments are other posterior uveitis complications commonly described [5,7]. The visual prognosis of these complications is particularly severe, and at least one-third of patients developing significant visual impairment. However, blindness, which was reported to involve about 18% of cases in the 1990s, is now quite rare, affecting only 5% of patients [1]. This is due to improvement in diagnosis, therapy and tight follow-up of these patients.

In this study, median duration of arthritis was about 13 years, progressing up to 35 years in one case. Uveitis occurred after a median interval of about one year after arthritis onset. This period of the time period was significantly longer than that reported in earlier studies [19,20,21], and this may reflect the current view to start uveitis screening immediately after JIA diagnosis.

Kotaniemi et al. [4] underline that rheumatologic/pediatric follow-up of JIA patients tends to stop before adulthood, whereas ophthalmologic follow-up is significantly extended over time. In accordance with these observations, we found in the present study that ocular and rheumatic disease were independently managed by the respective specialists. This may be due to the fact that our study was conducted in a tertiary referral center specializing in the treatment of eye inflammatory disease to which patients with severe eye disease were commonly referred. The relationship between uveitis and joint inflammation activity in our patients was not constant, with persistence of both conditions in some cases but not in others, as previously reported in other studies [21,22].

In contrast to the previous study [17], length of uveitis duration was related to early-onset arthritis, although the reasons for this observation cannot be definitely explained. Another finding from this study was that patients with a longer duration of uveitis had a later onset of uveitis compared with data reported by other authors [23]. This finding, however, is not statistically significant. This may be due to the small sample size analyzed.

Ocular complications of uveitis were observed in 12 patients (86%). This is in agreement with recent scientific literature reporting complicated uveitis in over half of the patients with JIA-U in long-term follow-up studies. Cataract is the most common complication observed, followed in frequency by band keratopathy and posterior synechiae [4,23]. In accordance with these studies, we observed that adults with JIA-U develop ocular complications more often than adults affected by chronic uveitis and, in particular, more than those affected by idiopathic non-infectious uveitis [24]. Posterior complications were also quite common in our patient population, including cystoid macular edema, which was found in 40% of cases. This percentage is significantly higher than previously described [23]. In one study, it has been suggested that macular edema may be an earlier JIA-U complication [17]. In this regard, we observed in the present study that macular edema was more common in adults than children. We are tempted to hypothesize that his finding may be explained by more extensive use of instrumental investigation in the adult population. In a previous study carried out at our Center, we highlight that those posterior complications are quite common, and investigation by OCT can detect subclinical macular edema, which is not identifiable by only the ophthalmoscopy.

The visual prognosis of JIA-U has greatly improved over the last 25 years. A comparison of data from studies of different periods [22,25] demonstrates that a significant visual impairment (≤0.95 LogMar) is now present in less than 50% of cases. Our study confirms this finding. In particular, visual acuity was good (≥0.95 LogMar) in all eyes affected by uveitis up to 15 years of follow-up. Treatment of uveitis in our study group was in agreement with the recommendations from the international committees, indicating the use of topical steroids as first-line medications in combination with mydriatic/cycloplegic drugs. The frequent use of medications for the control of IOP, as observed in our patients even in the absence of overt glaucoma, suggests the presence of transient disturbances of IOP regulation, possibly due to inflammation or as a consequence of steroid therapy. With regard to biologics, anti-TNF-alpha intravenous chimeric monoclonal antibody (mAb) infliximab was the most frequently used in our study population since it was the first approved drug for uveitis [26]. Subcutaneous humanized mAb adalimumab is now the anti-TNF-alpha biologic most commonly used for uveitis in clinical practice [27]. Lack of better visual prognosis as observed in patients treated with biologics may suffer from a bias because most severe cases were treated with biologics only after failure of first- or second-line conventional therapy. Although biological treatment appears to be a promising approach to controlling severe cases of ocular inflammation, further studies with a large sample are needed to evaluate the visual outcomes in this selected group of patients.

## 5. Conclusions

This study provides further evidence that JIA-U may progress from childhood to adulthood and that the most frequent complications of JIA-U in adult patients is the presence of posterior synechiae and cataract. Both conditions may require complex surgical treatments. Although the rate of complications of JIA-U has progressively declined over time as compared with 20 years ago due to extensive ophthalmological screening [23], much closer collaboration among rheumatologists, pediatricians and ophthalmologists is warranted to reduce the prevalence of these ocular complications. The use of new immunosuppressive drugs, including biologics and JAK inhibitors, is expected to lower further the progression rate and morbidity of JIA-U [28,29,30].

## Figures and Tables

**Table 1 jcm-11-02471-t001:** Demographic and clinical characteristics of patients.

Number of Patients	20
Number of eyes ongoing uveitis	35
Gender (% females)	85
Mean follow-up (years ± SD)	16 ± 7.7 (range 4–35)
Mean age at the end of follow-up (years ± SD)	23.4 ± 6.6 (range 16–38)
Median age at JIA onset (years ± SD)	6.15 ± 2.9 (range 2–10)
Median age at uveitis onset (years ± SD)	8.7 ± 4.7 (range 3–20)
Mean interval arthritis-uveitis onset (years ± SD)	1.3 ± 4.4 (range −5–10)
Bilateral uveitis (%)	75
ANA positive (%)	85
Uveitis before arthritis (%)	14.2
Median follow up (years ± SD)	16 ± 7.7 (range 4–35)
Ongoing arthritis (%)	70
Oligoarthritis (%)	95

**Table 2 jcm-11-02471-t002:** Incidence of ocular complications at baseline and during follow-up.

Complication	Total Affected Eyes	At Presentation	During Follow-Up ^a^	Eye/Year ^b^
Posterior synechiae	17 (48%)	12 (34%)	5/23 (21.7%)	0.009
Band keratopathy	18 (52%)	15 (42%)	3/20 (15%)	0.006
Cataract	18 (52%)	8 (22%)	10/27 (37%)	0.01
Elevated IOP	7 (35%)	2 (5.7%)	5/33 (15%)	0.006
Hypotony	2 (10%)	0 (0%)	2/35 (5.7%)	0.002
Cystoid macular edema	7 (35%)	2 (5.7%)	5/33 (15.5%)	0.006
Exudative retinal detachement	1 (5%)	0 (0%)	1/35 (2.8%)	0.001
Papillitis	4 (20%)	2 (5.7%)	2/33 (6%)	0.002
Epiretinal membrane	2 (10%)	0 (0%)	2/35 (5.7%)	0.002

^a^ Number of events/number of eyes. ^b^ Number of events expressed as eye/year during follow-up.

**Table 3 jcm-11-02471-t003:** Systemic treatments used in JIA-uveitis patients.

Drug	Patients
NSAID	6 (30%)
Corticosteroids	8 (40%)
Gold salts	1 (5%)
MTX	9 (45%)
Cyclosporine	3 (15%)
Azathioprine	1 (5%)
Infliximab	8 (40%)
Rituximab	1 (5%)
Adalimumab	3 (15%)
Abatacept	1 (5%)

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
