# Peer review of "Persistence of Juvenile Idiopathic Arthritis-Associated Uveitis in Adulthood: A Retrospective Study"

_jcm, 2022, doi:10.3390/jcm11092471_

Round 1

Reviewer 1 Report

It is always interesting to report such a series of patients with JIA and uveitis, especially its evolution at adult age. Nicely written, easy to read, I detected one fault in english (page 4 "median age of affected patients" instead of affects.

As the authors say "The visual prognosis JIA-Uhas greatly improved over the last 25 years." May be its is worth of interest to look if there is a difference between the DMARDS treated patients and the Biologic treatment group ?

In summary, some small modifcations.

Author Response

Rome, April 25th 2022

Dear Reviewer,

We firstly thank you for your suggestions and for the general appreciation of our manuscript. We corrected the fault in English you detected on page 4. English was revised throughout the manuscript and both spelling and grammar were carefully checked.

We answered to your question about the possible difference in the visual outcome between patients treated with DMARDS and those treated with Biologics adding the following sentence in the Results section: “We didn’t find any difference in visual prognosis of patients treated with csDMARDs alone or in combination with biologics”. We also added the sentence “Lack of better visual prognosis as observed in patients treated with biologics may suffer from a bias, because most severe cases were treated with biologics only after failure of first- or second-line conventional therapy. Although biological treatment appears to be a promising approach to control severe cases of ocular inflammation, further studies with large sample are needed to evaluate the visual outcomes in this selected group of patients”. It is therefore possible that treatment of JIA-U patients with biologics at an earlier phase of eye disease could lead to a better visual outcome than treatments with conventional DMARDs.

We hope that our responses have fulfilled all of your queries.

Kind regards,

Maria Pia Paroli, MD

Reviewer 2 Report

This paper offers a nice clinical study on outcome in JIA related uveitis.

Table 1 should be fixed. The  legend is OK but the table appears confusing with numbers which do not correspond to the legend.

In the materials and methods section:

A. Cataract: It would be nice to have the full scaled used for the grading of cataract opacification grade 1 and definition of cataract grading.

B. Macular thickening : Please define the range of OCT measurement that was considered as macular edema.

The authors should also mention as eye/year follow-up of patients and not only  the median follow-up of patients to give a better idea of complications observed. 

For medication please provide information about the type of steroid used or orbital floor injection.

Author Response

Rome, April 25th 2022

Dear Reviewer,

We firstly thank you for your interest in our manuscript and for your comments. You firstly observed that Table 1 appears confusing with number which do not correspond to the legend. The table has been rewritten and the numbers have been corrected in accordance with the results described in the text.

You also commented that it would be nice to have a full scaled used for grading cataract opacification grade 1 and definition of cataract grading. Cataract grading was performed in our study using an arbitrary measure system. However, we added the following sentence in the Discussion session to underline the importance of a grading system such as LOCS III for the assessment of lens opacity: “A potential limitation of our cataract grading was that it was not based on the Lens Opacities Classification System (LOCS) III. This is a method designed for classifying lens opacities that involves a standardized classification of nuclear color at the slit lamp and cortical and posterior subcapsular cataracts at the slit lamp with retro-illumination. However, LOCS III was introduced in 1993 and our study analyzed patients since 1990. It is desirable that further studies on JIA-U might take advantage of LOCS III to better assess lens opacity complications”. A reference on this topic has been added to the reference list.

You then asked to define the range of OCT measurement that was considered as macular edema. In this regard the following statement has been added in the Materials and Methods section: “Macular edema was defined as central macular thickness (CMT) greater than 300 µm with or without cyst formation as detected by clinical examination and/or by spectral domain OCT”. An additional reference has been also included.

You also commented that eye/year follow-up of patients should be mentioned to give a better idea of complication observed. These data have now been included in the Results section and in Table 2.

Finally, you asked to provide information about the type of steroid administered either parenterally or topically to our patients. Therefore, the type of steroids used has been specified throughout the text.   

We hope that our responses have fulfilled all of your queries.

Kind regards,

Maria Pia Paroli, MD